# Single-Cell Transcriptome Sequence Profiling on the Morphogenesis of Secondary Hair Follicles in Ordos Fine-Wool Sheep

**DOI:** 10.3390/ijms25010584

**Published:** 2024-01-02

**Authors:** Chenglan Li, Xue He, Yi Wu, Jianye Li, Rui Zhang, Xuejiao An, Yaojing Yue

**Affiliations:** 1Key Laboratory of Animal Genetics and Breeding on the Tibetan Plateau, Ministry of Agriculture and Rural Affairs, Lanzhou Institute of Husbandry and Pharmaceutical Sciences, Chinese Academy of Agricultural Sciences, Lanzhou 730050, China; lichenglan434@163.com (C.L.);; 2Sheep Breeding Engineering Technology Research Center, Chinese Academy of Agricultural Sciences, Lanzhou 730050, China

**Keywords:** ordos fine-wool sheep, secondary hair follicle morphogenesis, single-cell transcriptome sequencing, pseudo-temporal differentiation trajectory

## Abstract

The Ordos fine-wool sheep is a high-quality breed in China that produces superior natural textiles and raw materials such as wool and lamb meat. However, compared to the Australian Merino sheep, there is still a gap in terms of the wool fiber fineness and wool yield. The hair follicle is the main organ that controls the type of wool fiber, and the morphological changes in the secondary hair follicle are crucial in determining wool quality. However, the process and molecular mechanisms of hair follicle morphogenesis in Ordos fine-wool sheep are not yet clear. Therefore, analyzing the molecular mechanisms underlying the process of follicle formation is of great significance for improving the fiber diameter and wool production of Ordos fine-wool sheep. The differential expressed genes, *APOD*, *POSTN*, *KRT5*, and *KRT15*, which related to primary hair follicles and secondary hair follicles, were extracted from the dermal papillae. Based on pseudo-time analysis, the differentiation trajectories of dermal lineage cells and epidermal lineage cells in the Ordos fine-wool sheep were successfully constructed, providing a theoretical basis for breeding research in Ordos fine-wool sheep.

## 1. Introduction

Fine-wool sheep produce fine natural textile materials and sheep meat [1,2]. Since 1986, China’s Ordos has introduced Australian Merino rams and local Mongolian ewes for genetic improvement through the infusion of new favorable alleles. The three consecutive phases of breeding and improvement projects have gradually developed the Ordos fine-wool sheep, which can be used for wool, meat, and skin. The sheepskin of Ordos fine-wool sheep is known for its substantial thickness, strong elasticity, and excellent softness, making it the best raw material for leather products [3]. Wool fineness is the main factor affecting wool prices and is also the most important indicator for evaluating wool quality in the wool-spinning industry. Compared with superior wool-producing animal breeds such as Australian Merino sheep, there is still a significant gap in wool fineness and wool yield in Ordos fine-wool sheep [4]. There is an urgent need to research the molecular mechanisms of key economic traits such as fineness and wool yield and explore and control the major genes associated with the wool fiber diameter and wool yield. This will provide a theoretical basis and technical support for accelerating the breeding of Ordos fine-wool sheep.

The hair follicle, as an important skin appendage, is a complex and intricate organ specific to mammals, composed of multiple types of cells [5]. Its morphogenesis involves interactions between the epidermis and dermis, and the formation of the hair follicle structure is a highly complex process facilitated by the interaction of various types of cells [6]. The fur traits in economically valuable animals are mainly determined by the cells during the early stage of embryonic development when the hair follicles are initially formed. Therefore, hair follicle traits directly affect the yield and quality of the fur in these animals [7]. Hair follicles also have strong self-renewal abilities due to the presence of a group of stem cells. The growth of hair follicles and the activity of these stem cells are highly regulated by various signaling pathways. Therefore, the self-renewal ability of hair follicles not only contributes to hair growth and regeneration but also facilitates skin regeneration after injury. As a result, the hair follicle can serve as an important model for tissue regeneration and systems biology research [8], playing an irreplaceable role in the processes of skin function and regeneration. Therefore, understanding the developmental rules of hair follicles is of great importance in guiding the production of important economic traits in fine-wool sheep, such as the fiber diameter and length, as well as for guiding the production of down animals.

The initial morphological studies using the hematoxylineosin staining method (HE), gene knockout technology at the gene editing level, and now the rapidly evolving Omics technology have provided technical support for studying hair follicle morphogenesis. These advancements have made it possible for researchers to conduct multi-level and multi-dimensional studies in the field of hair follicle morphogenesis. Nie used HE staining technology to study the morphological structures of hair follicle development during the embryonic period of Tibetan sheep. They found that during the induction period (Stage 0), the epidermis was relatively thin, but the thickness of the epidermis significantly increased at Stage 1 [9]. In recent years, single-cell RNA sequencing (scRNA-seq) has been applied to various species, especially different human tissues, revealing changes and variations in gene expression among cells [10,11,12,13,14]. Currently, this technology is mainly applied in the field of skin wound repair and regeneration in humans or mice [15]. In addition, Wang applied scRNA-seq to identify and analyze differentially expressed genes in specific cells of straight and curly hair, revealing differentiation in sheep hair follicle development [16]. Some researchers used scRNA-seq to sequence goat fetuses and found correlations in gene expression at different development stages [17,18]. Many studies have successfully used scRNA-seq to reveal the molecular mechanism of mouse hair follicle development [19,20,21], suggesting that single-cell transcriptome sequencing can be an effective method in explaining the mechanism of hair follicle development. However, the understanding of the hair follicle development in fine-wool sheep is still limited. This study aimed to eliminate the influence of irrelevant variables such as gender on the results. Therefore, we selected only single male lambs for the study. Firstly, we applied the hematoxylineosin staining method to investigate the process of secondary hair follicle morphogenesis in the Ordos fine-wool sheep. Then, based on the scRNA-seq technology, we conducted single-cell sequencing of the hair follicle during the process of morphogenesis in Ordos fine-wool sheep and constructed a single-cell transcriptional atlas of secondary hair follicle development. Based on the clustering analysis results of single cells, we first identified the major cell types involved in the process of hair follicle morphogenesis in Ordos fine-wool sheep. Furthermore, we provided a detailed description of the transcriptomic profiles of different cell types at the single-cell level. This study enriched our understanding of the regulatory mechanisms underlying hair follicle morphogenesis in Ordos fine-wool sheep, aiming to provide theoretical support for the future molecular breeding of super-fine-wool sheep.

## 2. Results

### 2.1. Morphogenetic Identification of the Early Development of Wool Follicles in Ordos Fine-Wool Sheep

According to the HE staining results, morphological observations were conducted on the skin tissues of Ordos finewool sheep. In the initial stage of hair follicle morphogenesis, the E81 epidermis begins to undergo preliminary invagination and differentiates into the thickened E84 hair follicle placode, where a homogeneous cell structure layer can be clearly observed. By E87, the hair follicle further elongates and invaginates, and the inner root sheath (IRS) and outer root sheath (ORS) can be observed clearly, but mature dermal papilla (DP) has not yet formed. Further differentiation and maturation are still required. By E90, the DP structure undergoes differentiation in the bulb region at the base of the hair follicle, and the sweat gland structure of the primary hair follicle can be observed. At E93, the DP structure is surrounded by a matrix, and enlarged sebaceous gland cells have formed. By E110, mature primary and secondary hair follicles can be observed, with the primary hair follicle possessing sebaceous glands, sweat glands, and arrector pili muscle structures, while the secondary hair follicle only has sebaceous glands. Additionally, the primary hair follicles are thicker and more robust compared to the slender secondary hair follicles (Figure 1).

### 2.2. Identification and Molecular Characteristics of Major Cell Types

This study integrated quality-controlled data from two time periods, E93 and E110. After integration, a total of 21,342 transcriptome profiles and 31,588 genes were obtained. Initially, tSNE (t-Distributed Stochastic Neighbor Embedding) clustering analysis was performed on the data, resulting in the identification of 18 cell clusters (Figure 2a). The results showed that cells from the E93 period were mainly concentrated on the right side of the tSNE plot, while cells from the E110 period were mainly concentrated on the left side of the tSNE plot (Figure 2b,c), indicating a correlation between their distribution and developmental progression.

Firstly, a heat map of differentially expressed genes in each cell cluster was generated, and the results showed that although there were differences among different cell clusters, there was still some correlation between them (Figure 1). In this study, the specific expression genes of each cell cluster were compared with the marker genes identified in previous relevant studies to identify each cell cluster. The results showed that cell clusters 2, 5, and 9 expressed the dermal cell marker genes COL1A2 and COL1A1; cell clusters 5 and 9 expressed the fibroblast marker genes OGN and DCN; cell cluster 6 expressed the DP marker gene APOD; cell clusters 0, 1, 3, 4, 10, and 11 expressed the epidermal cell marker genes KRT15 and KRT5; cell cluster 10 expressed the hair shaft marker gene MSX1; and cell cluster 4 expressed the keratinocyte genes KRT1 and KRTDAP. Furthermore, cell cluster 11 expressed the SG marker gene MGST1; cell cluster 17 expressed the melanocyte markers TYRP1 and MLANA; cell clusters 8, 12, and 14 expressed the epidermal markers TPM2 and ACTA2; cell cluster 16 expressed the endothelial cell markers PECAM1 and KDR; and cell cluster 15 expressed the immune cell marker CTSS (Figure 3). The proportion of epidermal and fibroblast cells is the highest, and the number of dermal cells at the E110 stage is significantly reduced compared to that at the E93 stage, while the number of epidermal cells significantly increased (Figure 4). Through the above analysis, the main cell types involved in the morphogenesis of wool follicles in fine-wool sheep have been successfully identified.

This study compared the specific expression genes of each cell population using multiple visualization methods. The results showed that hypodermal cells highly expressed *COL1A2*, *COL1A1*, and *LUM*; fibroblasts expressed *OGN* and *DCN*; DP cells exhibited a high expression of *APOD*; funnel cells had a high expression of *CENPF*, *TOP2A*, *UBE2C*, *TPX2*, and *LRIG1*; epidermal cells showed a high expression of *SOX9*, *KRT15*, *KRT5*, and *KRT17*; keratinocytes expressed *KRT1, KRTDAP, KRT10*, and *SBSN*; sebaceous glands exhibited a high expression of *MGST1*; melanocytes highly expressed *TYRP1*, *MLANA*, and *DCT*; periderm cells showed a high expression of *TPM2* and *ACTA2*; immune cells exhibited a high expression of *CTS*, *RGS*, *FCER1A*, *LCP1*, *RGS10*, and *LTC4S*; and endothelial cells highly expressed *PECAM1* and *KDR*. The official names of the above genes are shown in Appendix A. The genes mentioned above are marker genes for their corresponding cell types, further identifying the specificity of each cell type (Figure 5).

### 2.3. The Construction of the Differentiation Trajectory of Dermal Lineage Cells during the Wool Follicle Morphogenesis Process

To study the process of morphogenesis in the wool follicles of the Ordos fine-wool sheep, the main cell types involved in the process were identified, and a pseudo-temporal trajectory was constructed using Monocle. The initially extracted dermal lineage cells consisted of dermal cells, fibroblasts (Fb), and dermal papilla (DP), and a pseudo-temporal trajectory was constructed (Figure 6b). In order to further investigate the specialization of different cell types within the dermal lineage during their differentiation process, differential gene analysis was performed on different branch points along the pseudo-temporal trajectory, revealing the dynamic changes in gene expression. By establishing the pseudo-temporal trajectory using Monocle, two branch points were identified in the dermal lineage cells (Figure 6a). According to relevant studies, the expression level of *COL1A1* decreases during the differentiation process of dermal lineage cells (Figure 6d). Additionally, E93 is mainly concentrated in Stages 1 and 2 (Figure 6c). Therefore, Stage 1 is selected as the starting point for the trajectory analysis (Figure 6e). *LFE1*, *PRDM1*, and *RSPO3*, as marker genes for the DC (Dermal condensate) precursor, DC1, and DC2 stages in the DC differentiation process, exhibit expression patterns consistent with mice (Figure 6f). It is speculated that there is a complete DC cell differentiation process in the proposed temporal trajectory of dermal lineage cells. To validate this hypothesis, a detailed analysis will be conducted on each branching point.

First, we constructed a gene expression trend map for branch point 1 and divided it into five gene sets based on the expression patterns. We then performed enrichment analysis on the related genes (Figure 7a). In branch point 1, during Stage 1, the marker genes *OGN*, *DCN*, and *VIM* were highly expressed in gene sets 4 and 5, and there was enrichment in the pathways related to fibroblast proliferation, cell cycle regulation, and Wnt signaling. This indicates that, during this stage, dermal progenitor cells undergo proliferation and differentiation into fibroblasts. Cell fate 2 exhibits the differential expression of epidermal-related marker genes, such as KRT1, KRTDAP, and KRT10, from gene set 3. It is also significantly enriched in cell adhesion and in cell-cycle-related pathways (Figure 7a,b). This suggests that, during this stage, some cells from the hypodermal cell are engulfed by the epidermis, leading to their differentiation into epidermal-related cell types such as keratinocytes and SG cells. It is hypothesized that the differentiation trajectory of these cells represents a partial transformation of hypodermal cells engulfed by the epidermis, but they have not completely transitioned into epidermal cells. In Stage 2 of Branch 2, fibroblast marker genes, such as *COL1A1*, *COL14A1*, and *MEST*, showed a decreasing trend along the pseudo-temporal trajectory, and they were enriched in cell adhesion and Wnt/β-catenin signaling. This further demonstrates the differentiation of fibroblasts towards dermal condensate (DC). In Cell fate 2, DP marker genes, including *TOP2A*, *CXCR4*, and *FGFR1*, exhibited an increasing trend along the pseudo-temporal trajectory, and they were enriched in cell proliferation, the negative regulation of Notch signaling, and the negative regulation of Wnt signaling. Fibroblast marker genes, such as Cell fate 1, *DCN*, *OGN*, *CRABP1*, *ELN*, *PDGFRA*, and *VIN*, showed an upward trend along the pseudo-temporal trajectory, and they were enriched in the negative regulation of cell proliferation, BMP signaling, and the positive regulation of Wnt/β-catenin signaling pathways (Figure 7c,d). These findings provide preliminary evidence of the significant differences in the morphogenesis process of DP cells in primary and secondary hair follicles.

### 2.4. The Construction of the Differentiation Trajectory of Epidermal Lineage Cells during the Wool Follicle Morphogenesis Process

To analyze the process of epidermal lineage cell differentiation during hair follicle morphogenesis, this study selected epidermal lineage cells, including epidermal cells, keratinocytes, sebocytes, and hair shaft cells, to construct a quasi-temporal trajectory. Two branching points were identified in the quasi-temporal trajectory (Figure 8a–c). First, *CTNNB1*, a classic marker gene for the basal layer of the epidermis, showed high expression levels along the trajectory, consistent with the differentiation pattern. Using Stage 1 as the starting point of the trajectory, it was further observed that *KRT25*, *LUM*, and *POSTN* exhibited an increasing trend during epidermal differentiation, while *SBSN* showed a decreasing trend (Figure 8d–f).

Branch 2, Stage 1 is enriched in terms of translation and the related pathways, preparing for subsequent differentiation processes. Then, Stage 2 is enriched in pathways such as cell proliferation, with a high expression of keratinocyte marker genes such as *KRTDAP*, *KRT1*, *SBSN*, and *KRT10*. Stage 3 is enriched in translation- and cell adhesion-related pathways, preparing for subsequent differentiation processes related to secondary hair follicles (Figure 9a,b). In Branch 1, Cell fate 1 describes the differentiation process of keratinocytes and hair shafts associated with secondary hair follicles. *DLX3*, as a marker gene promoting the differentiation of secondary hair follicles, is upregulated in Branch 1, indicating a close relationship between Branch 1 and secondary hair follicle differentiation. Stage 3 is enriched in pathways related to keratinocyte differentiation. Stage 5 highly expresses marker genes of hair follicle stem cells and fibroblasts, such as *KRT14*, *KRT10*, *SPARC*, *SFRP1*, *LHX2*, *COL14A1*, and *COL7A1*. It enriches pathways related to keratinocyte formation and cell migration. On the other hand, Cell fate 2 describes the differentiation process of muscle cells in the epidermis, including the formation of hair muscles. In Stage 4, marker genes of fibroblasts and epidermal cells, such as *COL1A1*, *VIM*, *KRT15*, and *KRT17*, are highly expressed, enriching pathways related to fibrous tissue development (Figure 9c,d).

A GO enrichment analysis was conducted to analyze the differential genes related to the differentiation process of two differentiating stages of keratinocytes in branches 2 and 1. The enriched keratinocyte differentiation-related pathways were then analyzed for their interactions with the WNT/β-CATENIN, BMP, translation, and cell adhesion signaling pathways. The results showed that the differentiation of keratinocytes in branch 2 is closely associated with BMP signaling, translation, and cell adhesion pathways (Figure 10a). On the other hand, the differentiation of keratinocytes in branch 1 is associated with WNT/β-CATENIN signaling and changes in cell morphology but not clearly linked to transcription-related pathways. This further indicates that there are certain differences in the molecular characteristics of keratinocytes at different differentiation stages (Figure 10b).

### 2.5. Fluorescent Immunohistochemical Validation

We randomly selected three marker genes and validated them using the fluorescence immunohistochemistry method. The results indicate that KRT15, as a marker gene for the epidermis, exhibits significant differential expression in the hair follicle epidermis (Figure 11a). Spatial information reveals the differential expression of KRT14 in the hair shaft cells (Figure 11b). TOP2A shows high expression in the dermal papilla (DP) (Figure 11c). In a word, KRT15 is validated at the epidermis of the Hair follicle, TOP2A is validated at the Dermal papilla (Stage 2), and KRT14 is validated at the Hair shaft (Stage 5). The localization results are consistent with the cellular identification results, indicating the accuracy and reliability of the data analysis.

## 3. Discussion

### 3.1. Identification of Major Cell Types

In previous studies on Cashmere goats, it was found that their hairs form a type of heterogeneous coat. The primary hair follicles occur at an earlier stage (E60), they are long and robust, and the bulbous part of the hair follicle is large. They grow as single strands in the skin and have a deeper distribution. Primary hair follicles generally produce coarse hair with a medullated fiber. The secondary hair follicles occur at a later stage (E80). They are short and slender, and the bulbous part of the hair follicle is small. They have a shallower distribution in the skin and are responsible for producing non-medullated cashmere fibers [22]. The results of this study indicate that, in the lateral skin tissue of E93 Ordos fine-wool sheep, the epidermal layer is structurally composed of a uniform layer of cells, forming the structure of the hair follicle matrix. The dermis has undergone invagination and differentiation into mature hair follicle structures, with enlarged sebaceous gland cells and elongated sweat glands and arrector pili muscles formed in the upper part of the hair follicle. It can be seen that, at this time, the hair follicles are mainly composed of primary hair follicles. Relevant studies on Chinese Super-Fine-Wool Sheep have found that secondary hair follicles begin to develop at E87 and, by E138, most of the primary hair follicles and some secondary hair follicles have matured [23]. Research also indicates that, during the period of E80–85, the secondary hair follicles start to form in the embryonic skin of the Subo Merino sheep, while a large number of primary hair follicles are also formed during this period [24]. During the embryonic stage of E110 in the Ordos fine-wool sheep, the lateral skin has already differentiated into mature primary hair follicles, which possess structures such as sebaceous glands, sweat glands, and arrector pili muscles. Additionally, secondary hair follicles with only sebaceous glands have also differentiated. These structures exhibit the morphological characteristics of primary and secondary hair follicles. It can be observed that the differentiation of secondary hair follicles in Ordos fine-wool sheep occurs relatively late and lasts for a longer duration. The differences between the research results mentioned above and the results of this study regarding high-quality wool-producing animals suggest that there are variations in the morphogenesis process of secondary hair follicles among different breeds of high-quality wool-producing animals. It is speculated that these differences arise due to variations in the timing and process of hair follicle morphogenesis across different species.

Through the HE staining technique and morphological observation, it was found that there were obvious mature primary hair follicles in the E93 stage, but secondary hair follicles did not appear. In the E110 stage, both distinct primary and secondary hair follicles were differentiated. These two stages can be used for studying the molecular characteristics of secondary hair follicles in fine-wool sheep. The differences in the E93 and E110 stages of fetal age in Ordos fine-wool sheep can be utilized for studying the morphogenesis of secondary hair follicles using scRNA-Seq.

### 3.2. Construction of Differentiation Trajectories of Dermal Lineage Cells

The hair follicle morphogenesis process involves complex interactions between signaling pathways such as Wnt, Hedgehog, Notch, and Bone Morphogenetic Protein (BMP). This study investigated the differentiation process of dermal lineage cells from fibroblasts (Fb) to dermal papilla cells (DP), using a pseudo-temporal analysis trajectory. The starting point of the pseudo-temporal trajectory, referred to as stage 1, showed some cell accumulation, indicating that some Fb cells did not undergo transformation into dermal condensate (DC) cells and did not participate in the process of hair follicle morphogenesis. However, some Fb cells underwent condensation and progressed through three stages: DC precursor, DC1, and DC2, eventually differentiating into mature DP cells. This process is induced by the BMP, Notch, and Wnt/β-catenin signaling pathways. In the early stage of hair follicle morphogenesis, BMPs secrete signaling molecules of the TGF-β superfamily, which exert their biological activity by binding to specific receptors [25,26,27,28]. As a multifunctional regulatory factor, it can control cell proliferation or apoptosis in various organs [29,30,31]. However, during hair follicle morphogenesis, the BMP signaling pathway acts through inhibitory regulation. It can also inhibit hair follicle growth by regulating the activation and amplification of hair matrix precursor cells, playing a role in hair follicle morphogenesis, regeneration, and the control of the hair follicle cycle after maturation [29]. Relevant studies have shown that the WNT/β-CATENIN signaling pathway also plays an important regulatory role in the early morphogenesis of hair follicles [32]. The conditional knockout of β-catenin in DC cells also prevents the formation of Pc structures, resulting in abnormal hair follicle differentiation. Knocking out the key mediator protein β-catenin of the Wnt/β-catenin signaling pathway in hair follicles leads to significantly shorter and thinner hair, morphological changes, reduced matrix cell proliferation rates, and altered gene expression patterns in hair shafts and also affects hair follicle growth [33]. The Notch signaling pathway plays an important role in the morphogenesis of hair follicles by controlling the fate of various cells [34]. During the morphogenesis of hair follicles, the basement membrane begins to proliferate, dermal progenitor cells condense, and keratinocytes start to form by inhibiting BMP signals. This inhibition facilitates the smooth progression of hair follicle morphogenesis [35]. In the dermis, noggin mediates the inhibition of BMP. Relevant studies have shown that Notch also antagonizes Wnt signal transduction. In the dermis, the Notch/RBP-Jk signaling pathway activates the expression of Wnt5a, which is facilitated by the binding of Notch1 to the RBP-Jk binding site in the promoter region. Wnt5a mediates Notch signal transduction by promoting the expression of the FoxN1 gene [36]. The above research results not only further demonstrate the accuracy of cell type identification in this study but also outline the differentiation process of dermal lineage cells and the interactions between the pathways involved in hair follicle morphogenesis. Additionally, this study provides insights into the expression changes in the expression of multiple genes during this differentiation process.

### 3.3. Construction of the Differentiation Trajectory of Epidermal Progenitor Cells

This study utilizes a pseudo-time analysis trajectory to characterize the intricate differentiation process of epidermal progenitor cells relative to dermal progenitor cells. In comparison to the pseudo-time differentiation process of dermal progenitor cells, the differentiation process of epidermal progenitor cells is more complex. First, the expression of marker genes on the substrate is significantly abundant during the E93 and E110 stages, thereby promoting the morphogenesis of hair follicles. Branch 2 primarily aggregates the early cornified cells associated with the primary hair follicle at the E93 stage. Branch 1 describes the differentiation process of the arrector pili muscle of the primary hair follicle, as well as the differentiation process of the cornified cells and hair shafts associated with secondary hair follicles. Branch 2, as the branch closest to the starting point of the track, exhibits an active differentiation state, preparing for subsequent differentiation. The highly differentiated branch associated with the primary hair follicle demonstrates a state of self-renewal. On the other hand, Branch 1 describes the differentiation process of the keratinocytes and hair shafts associated with the secondary hair follicle, as well as the differentiation process of muscle cells in the epidermis, such as the arrector pili muscle. This further illustrates the asynchronous development of the hair follicle. As the arrector pili muscle and a specific part of the primary hair follicle, it undergoes differentiation during the morphogenesis of the secondary hair follicle. This indicates a connection between the differentiation process of the arrector pili muscle and the keratinocyte differentiation process of the secondary hair follicle. Protein interaction analysis was conducted separately on the differentiation processes of primary and secondary hair follicles and described the differentiation of keratinocytes at different stages. The analysis revealed significant differences in the keratinocyte differentiation between the two stages. The differentiation of keratinocytes in primary hair follicles is closely associated with BMP signaling, translation, and cell adhesion pathways. On the other hand, the differentiation process of cornified cells in secondary hair follicles is relatively complex and is associated with WNT/β-CATENIN signaling and cellular morphological changes. However, there is no apparent association with transcription-related pathways. The BMP signaling pathway regulates the proliferation and differentiation of hair follicle matrix precursor cells, con-trolling the formation of hair follicle morphology and the hair follicle cycle defense [29]. Studies have shown that a high expression of WNT/β-catenin signaling promotes the formation of the hair follicle bulge and plays a regulatory role in the hair follicle morphogenesis and reformation processes. Additionally, the Wnt signal can regulate hair shaft differentiation through the mediation of BMPRIA signaling. In progenitor cells, BMPRIA activates hair-follicle-specific keratins through LEF1 and β-catenin, resulting in the generation of hair shafts [37]. Notch is involved in the morphogenesis of hair follicles through three mechanisms: lateral inhibition, boundary formation, and lineage determination. The fate of cells depends on the induction of Notch-related regulatory factors, which regulate differentiation and promote boundary formation by altering the adhesive properties of cornified cells [38]. Notch induces differentiation by suppressing the expression of p63. Notch interacts with Wnt in epidermal lineage cells to regulate cell adhesion and determine the cell position [39]. Therefore, it is speculated that the involvement of the Notch signaling pathway in the morphogenesis of primary hair-follicle-related cornified cells is indicated by the relevant pathways such as cell adhesion. It is also speculated that the induction of the Wnt signaling pathway, which is enriched during the secondary hair follicle period, differs from that of the primary hair follicle, requiring the involvement of multiple and more complex pathways.

## 4. Materials and Methods

### 4.1. Experimental Animals

With the approval of the Animal Ethics Committee of the Lanzhou Institute of Husbandry and Pharmaceutical Sciences of the Chinese Academy of Agricultural Sciences (Approval No: NKMYD202105), under the same feeding level and management conditions, artificial insemination was performed on 18 healthy 5-year-old Ordos fine-wool ewes with similar body weights and parities (three times) using semen from a healthy 4-year-old Ordos fine-wool ram. The days of artificial insemination were considered as Day 0. On Days 81, 84, 87, 90, 93, and 110 after insemination, three fetal lambs (all single male lambs with similar body weights) were collected via cesarean section. Two 2 cm × 2 cm skin samples were taken from the lateral aspect of each fetal lamb (posterior to the scapula) for an HE staining test. Based on the morphological identification of hair follicles stained with HE, skin samples from three fetal lambs were mixed at two time points, E93 and E110, respectively, for subsequent sequencing experiments.

### 4.2. Main Reagents and Instruments

The main reagents used in this study include polyformaldehyde (produced by Biyun Tian Biotechnology Co., Ltd.), anhydrous ethanol (produced by Nanjing Shengqinghe Chemical Co., Ltd., Nanjing, China), xylene (produced by Merck Life Science Co., Ltd., Shanghai, China), safranin dye (produced by Merck Life Science Co., Ltd.), eosin staining solution (produced by Biyun Tian Biotechnology Co., Ltd., Shanghai, China), Chromium™ Single Cell 3′ Library & Gel Bead Kit v3 (produced by 10× Genomics), Chromium™ Single Cell A Chip Kit (produced by 10× Genomics), Chromium i7 Multiplex Kit (produced by 10× Genomics), Dynabeads™ MyOne™ SILANE PN-2000048 (produced by 10× Genomics), SPRIselect Reagent Kit (produced by Beckman Coulter), Collagenase IV (produced by Sigma), and Trypan Blue (produced by Sigma). The antibody information used in this chapter is provided in the Appendix A.

Main instruments: XSP-13CA Biological Microscope (Shanghai Optical Instrument Factory, Shanghai, China), LEICA EG2265 Paraffin Slicer (Leica Microsystems (Shanghai, China) Trading Co., Ltd.), E0997 Slicing Blade (Leica Microsystems (Shanghai, China) Trading Co., Ltd.), Chromium Controller (10× Genomics), Agilent 2100 Bioanalyzer (Agilent), Dell T640 Tower Server (Dell), Cell Counter (Biorad), PCR Instrument (Hangzhou Jingge Scientific Instrument Co., Ltd., Hangzhou, China), Vortex-Genie 2 (Vortex-Genie), and 40 μm Cell Sieve (BD Falcon). The instruments and their respective companies are listed in the Appendix A.

### 4.3. Experimental Methods

#### 4.3.1. Observation of HE Staining

The separated skin tissue was fixed with 4% paraformaldehyde at 4 °C. After overnight fixation, the fixed tissue was dehydrated in an ethanol solution and then incubated in xylene for 30 min. The samples were embedded in paraffin blocks. The paraffin-embedded tissue was sliced into sections with a thickness of 5–7 μm using a paraffin microtome. The sections were spread in a water bath maintained at 45 °C, and the samples were lifted, unfolded, and adhered to glass slides. The samples were air-dried and stored in a drying oven at 45 °C. The dried glass slides were then dewaxed in xylene for 30 min, followed by rehydration in ethanol. After rehydration, the samples were stained with Hematoxylin and Eosin (HE) for 7 min, followed by rinsing twice with distilled water for 5 min each time. After rinsing with a 1% HCl (*v*/*v*) ethanol solution for 3–5 s, we immediately rinsed the glass slide with 45 °C water for 5 min for dehydration. We stained the glass slide with a 1% hydrochloric-acid–ethanol solution and further rinsed it with anhydrous ethanol for 10 min. We fixed the glass slide with a neutral resin mounting medium for observation and photography under an optical microscope.

#### 4.3.2. scRNA-Seq

We selected three fetal sheep skin samples from the E93 Stage and three samples from the E110 Stage, respectively, and put them together for cell suspension sequencing. The skin tissue fragments were washed three times with DMEM/F12 culture medium, and the surface-adherent blood stains were scraped off with forceps. Type IV collagenase digestion was performed at 37 °C for 30 min, followed by 0.25% trypsin digestion for 15 min. Filtering through a 40 μm cell strainer was conducted twice, followed by centrifugation at 1200 r/min for 5 min. Washing with PBS containing 0.04% BSA was carried out three times, after which the samples were placed on ice for later use. Prior to conducting subsequent experiments, it is necessary to assess the viability of the prepared single cells, with a cell viability greater than 80% required (trypan blue staining method). The cell concentration is to be adjusted to be between 700 and 1200/μL based on the counting results, with a volume of no less than 50 μL. Equal volumes of single-cell suspensions from three concurrent skin samples are to be mixed for cell capture. mRNA enriched with polyA tails is obtained through oligo(dT) magnetic bead enrichment. Subsequently, the obtained mRNA is fragmented using divalent cations in the Fragmentation Buffer. Using the fragmented mRNA as a template and oligonucleotides as primers, the first strand of cDNA is synthesized in the M-MuLV reverse transcriptase system. Then, the RNA chain is degraded by RNaseH, and the second strand of cDNA is synthesized using dNTPs as substrates in the DNA polymerase I system. The purified double-stranded cDNA is subjected to end repair, A-tailing, and sequencing adapter ligation. PCR amplification is performed on 370–420 bp cDNA fragments selected using AMPure XP beads, followed by the purification of the PCR products using AMPure XP beads again, resulting in the final library.

After adding four fluorescently labeled dNTPs, DNA polymerases, and primers to the sequencing sample, amplification is performed. During the extension of each complementary strand in a sequencing cluster, the incorporation of one fluorescently labeled dNTP released a corresponding fluorescence signal. The sequencer captured the fluorescence signals, and the software (AnalysisPipeline 5.11.4) converted the light signals into sequencing peaks, thereby obtaining the sequence information of the target fragment.

#### 4.3.3. Quality Control for 10× Sequencing Data

According to the official guidelines of 10× Genomics (https://www.10xgenomics.com/cn/, accessed on 15 April 2023), the raw sequencing files obtained are processed using the standard software, CellRanger v2.2.0. The generated raw barcode calling files are converted to fastq files using the Cellranger mkfastq function. After standard CellRanger counting, the generated gene expression matrix file is analyzed using the package Seurat v2.3.4, following the official user guide (https://satijalab.org/Seurat/vignettes.html, accessed on 1 June 2023). To perform quality control, the filtering cells function is used to filter out cells with fewer than 200 detected expressed genes and genes expressed in fewer than three cells. The “FilterCells” function is then used to remove these cells.

This study aimed to analyze the downstream analysis of each sample based on the information from the single-cell transcriptome. The data quality of each sample was as follows: the barcode detection efficiency of both samples was higher than 95%, and the genome alignment rate of the reads was greater than 85%. Among them, the E93 sample detected 12,929 pieces of transcriptome information and 19,997 genes, while the E110 sample detected 18,892 pieces of transcriptome information and 20,174 genes (Appendix A).

#### 4.3.4. t-SNE Clustering

After normalizing the data, the variable genes for each dataset were calculated, and downstream clustering analysis was performed. After standardization and data scaling, tSNE was used for clustering, and different cell clusters were partitioned using the FindClusters function. Then, the differential gene expression was compared between different cell clusters using the FindAllMarkers function. After obtaining the differential gene data, the Seurat plotting functions were used to visualize the differential genes, including using the FeaturePlot function to plot the distribution of gene expression in cells, using the DotPlot function to create a bubble plot of marker gene expression, and using the DimPlot function to plot the tSNE graph, with relabeled cell groups identified based on known conservative marker genes.

#### 4.3.5. Construction of Differentiation Trajectories for Dermal Lineage Cells and Epidermal Lineage Cells

Analysis was performed using the Monocle 2 and 3 software packages in the R platform based on the online tutorial (http://cole-trapnell-lab.github.io/monocle-release/docs/, accessed on 20 November 2023). Cells corresponding to the dermal lineage and epidermal lineage were selected from the Seurat object as the study objects for the Monocle analysis. Cell arrangement and trajectory construction were completed based on relevant genes. The BEAM function was utilized to analyze differentially expressed genes between cell branches, and the plot_genes_branched_heatmap function was used to generate expression heatmaps of branch-specific genes. Different gene sets were obtained through K-means clustering calculation.

#### 4.3.6. GO Functional Enrichment Analysis of Differentially Expressed Genes

We used Metascape Database (https://david.ncifcrf.gov/summary.jsp, accessed on 20 November 2023) for the gene ontology (GO) enrichment analysis of differentially expressed genes and visualized the enriched biological processes pathways using the R package ggplot2 in the R platform (https://ggplot2.tidyverse.org/, accessed on 20 November 2023).

#### 4.3.7. Immunohistochemistry (IHC) for Fluorescence Detection

The paraffin-embedded tissue slices were placed in a 60 °C oven for 2 h to prevent subsequent detachment. After baking, the following steps were performed for dewaxing and rehydration: Xylene I for 15 min, Xylene II for 15 min, 100% ethanol for 5 min, 90% ethanol for 5 min, 80% ethanol for 5 min, 70% ethanol for 5 min, 50% ethanol for 5 min, and PBS solution for 10 min. Antigen retrieval was performed using sodium citrate antigen retrieval solution (a solution of 3% trisodium citrate and 0.4% citric acid in ultrapure water). The antigen retrieval solution should be preheated in boiling water. After the slides have cooled to room temperature, the slides were transferred to the antigen retrieval solution and incubated for 10 min. After incubation, the slides continued to cool in the antigen retrieval solution until they reached room temperature. Then, the slides were transferred to a humidified box, and approximately 50 μL of the blocking solution (3% BSA and 10% donkey serum in TBS solution) was added to the slides. The slides were allowed to be blocked at room temperature for 30 min. After blocking, the primary antibody diluted in the blocking solution was added according to the recommended concentration in the antibody datasheet. The slides were covered with a coverslip after adding the primary antibody, and the humidified box was transferred for either overnight incubation at 4 °C or incubation at 37 °C for 2 h. After the primary antibody incubation was complete, the slides were removed and rinsed three times with TBST solution for 10 min each time to remove any unbound primary antibody. Then, the slides were placed back into a humidified chamber, and the pre-diluted secondary antibody was added. The slides were incubated at 37 °C for 1 h. Once the secondary antibody incubation was complete, the slides were rinsed three times with TBST solution. Then, the slides were transferred to a horizontal humidified chamber, and 30 μL of DAPI was added for nuclear staining. After staining, the slides were washed three times with PBS, and then 10 μL of antifade mounting medium was added. The slides were covered with a coverslip and stored in a light-protected environment.

## 5. Conclusions

This study observed histological sections of skin tissue from developing Ordos fine-wool sheep and found that there were obvious mature primary hair follicles at the E93 stage. However, no secondary hair follicles appeared. At the E110 stage, both primary and secondary hair follicles were differentiated. These two stages can be applied in studying the molecular characteristics of secondary hair follicles in fine-wool sheep. Based on pseudo-time analysis, researchers have successfully constructed the differentiation trajectories of dermal progenitor cells and epidermal progenitor cells. This has enabled the elucidation of the process by which dermal progenitor cells differentiate from fibroblasts (Fb) to dermal papilla (DP), as well as the differentiation process of keratinocytes, hair shafts, and arrector pili muscle cells within the epidermis that are associated with primary and secondary hair follicles. By conducting protein interaction analyses separately for the differentiation processes of keratinocytes in the primary and secondary hair follicles at different stages, significant differences in the differentiation of keratinocytes between these two stages were discovered.

## Figures and Tables

**Figure 1 ijms-25-00584-f001:**
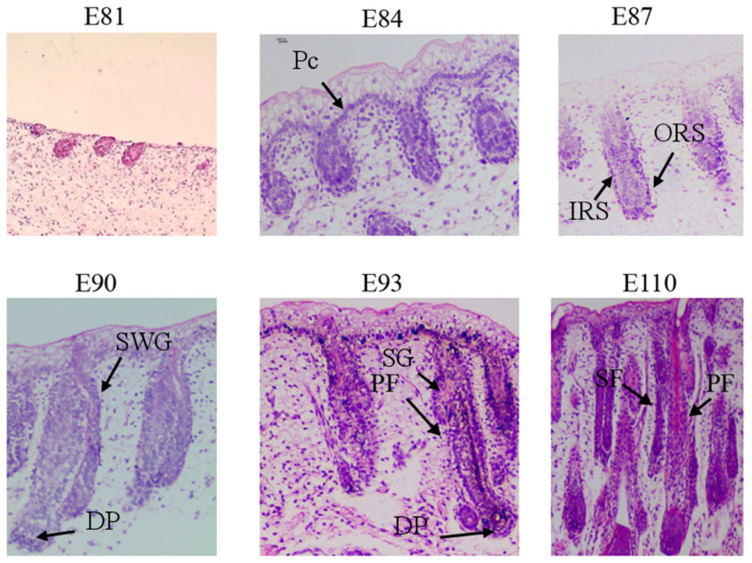
Morphogenesis of hair follicles in Ordos fine-wool sheep (HE staining, 400×). E81, E84, E87, E90, E93, and 110: Days 81, E84, E87, E90, E93, and 110 of embryo age; Pc: Placode; IRS: Inner root sheath; ORS: Outer root sheath; SWG: Sweat gland; DP: Dermal papilla; SG: Sebaceous gland; PF: Primary follicle; SF: Secondary follicle.

**Figure 2 ijms-25-00584-f002:**
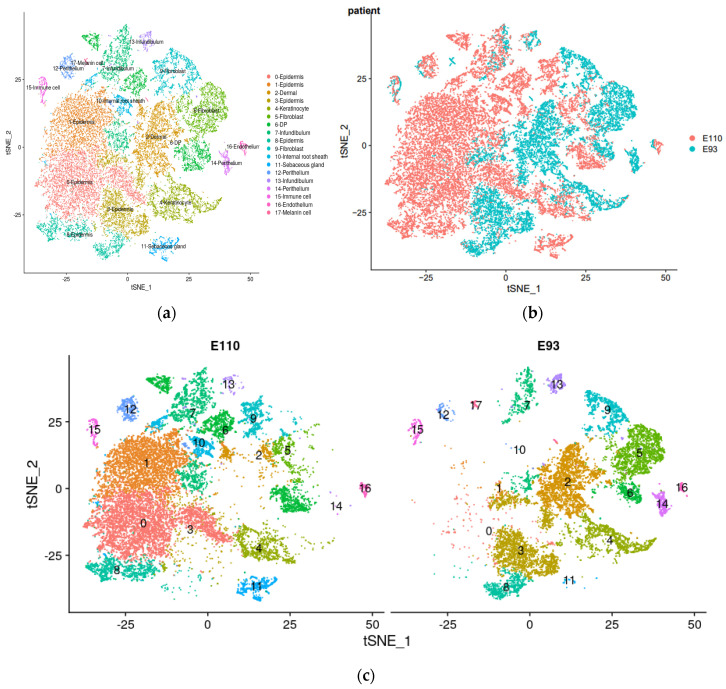
tSNE clustering results. (**a**) Information on subgroups in tSNE clustering results; (**b**) information on sample distribution in tSNE clustering results; (**c**) information on the distribution of each cell group in different samples in tSNE clustering results.

**Figure 3 ijms-25-00584-f003:**
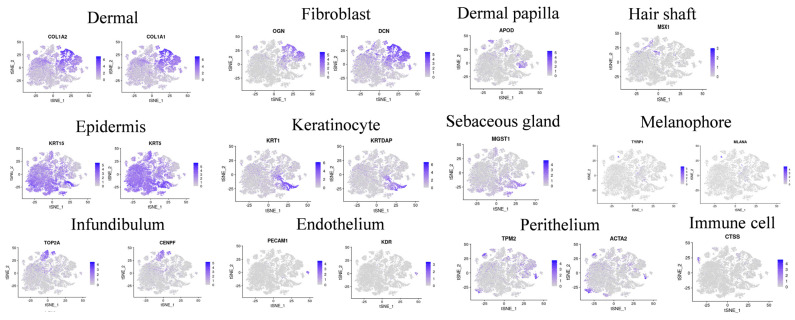
Expression of different types of cell marker genes in tSNE.

**Figure 4 ijms-25-00584-f004:**
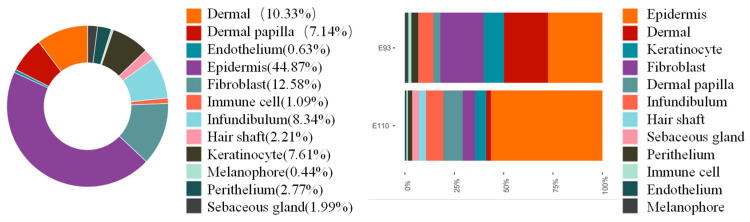
Comparative analysis of cell percentage by cell type.

**Figure 5 ijms-25-00584-f005:**
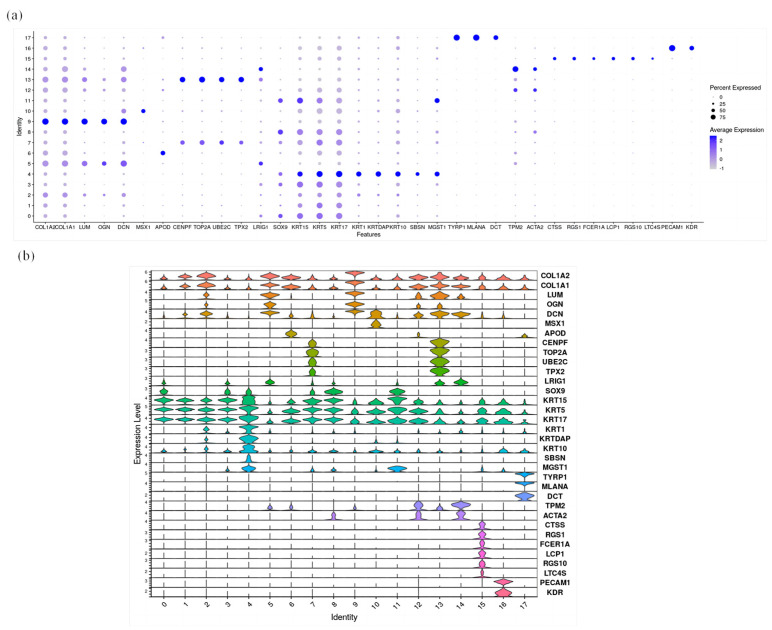
Comparative analysis of differential genes in different cell populations. (**a**) Bubble chart of the gene expression characteristics of each cell population; (**b**) violin diagram of the gene expression characteristics of each cell population.

**Figure 6 ijms-25-00584-f006:**
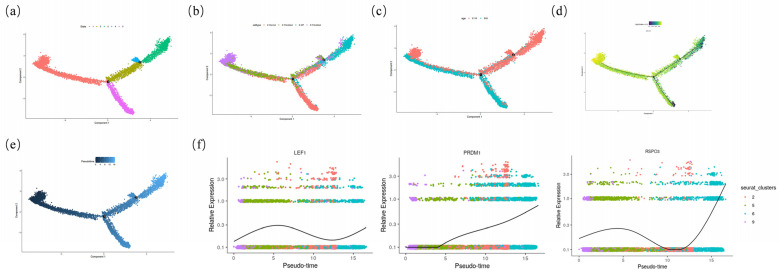
Construction of the dermal lineage cell pseudo-temporal trajectory. (**a**) Construction of the pseudo-temporal differentiation trajectory of dermal lineage cells; (**b**) construction of quasi-sequential trajectories based on cell populations; (**c**) sample-based quasi-temporal trajectory construction; (**d**) expression of *COL1A1* on the trajectory; (**e**) the starting point selection of the quasi-sequential trajectory; (**f**) expression of genes in different Stages of DC cells in fine-wool sheep.

**Figure 7 ijms-25-00584-f007:**
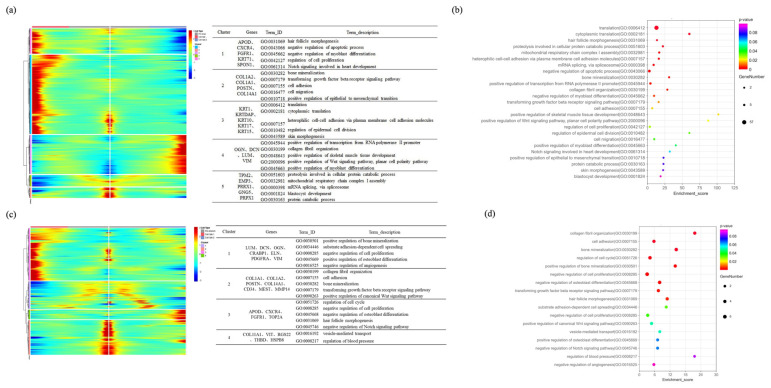
Dynamic changes in gene expression during cell differentiation dermal lineage. (**a**) Gene expression and GO enrichment analysis of differential genes in branch 1; (**b**) branch 1 differential gene GO enrichment analysis bubble diagram; (**c**) gene expression and GO enrichment analysis of differential genes in branch 2; (**d**) branch 2 differential gene GO enrichment analysis bubble diagram.

**Figure 8 ijms-25-00584-f008:**
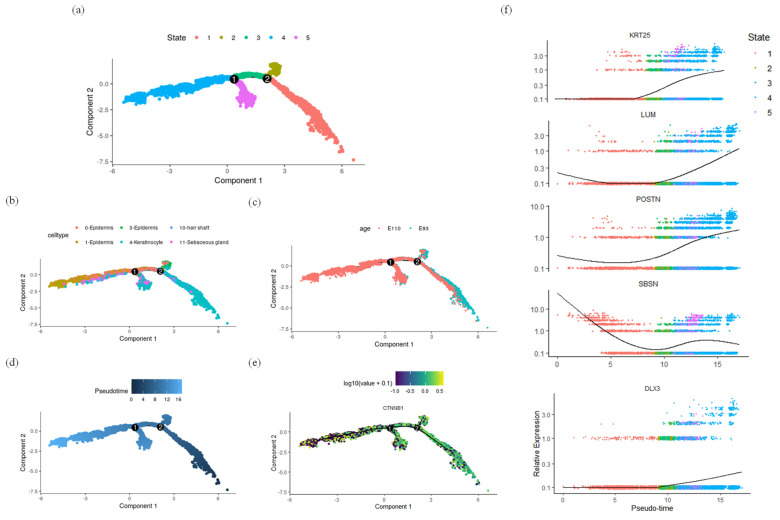
Construction of the epidermal lineage cell pseudo-temporal trajectory. (**a**) Construction of the pseudo-temporal differentiation trajectory of epidermal lineage cells; (**b**) construction of quasi-sequential trajectories based on cell populations; (**c**) sample-based quasi-timing trajectory construction; (**d**) the starting point selection of the quasi-sequential trajectory; (**e**) expression of CTNNB1 on the trajectory; (**f**) expression changes in differential genes with quasi-temporal trajectories.

**Figure 9 ijms-25-00584-f009:**
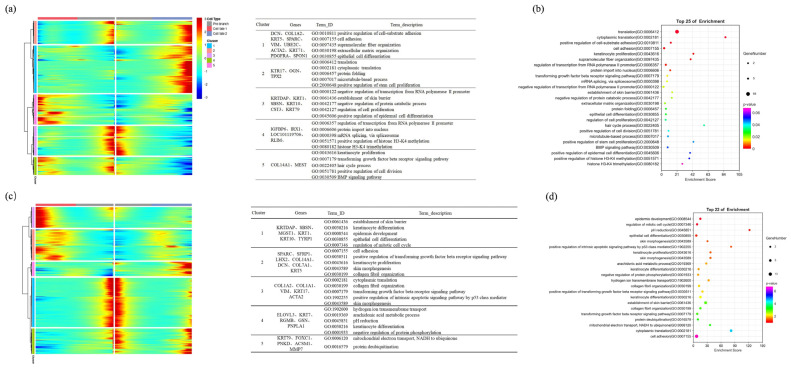
Dynamic expression of gene expression during the differentiation epidermal. (**a**) Gene expression and GO enrichment analysis of differential genes in branch 2; (**b**) branch 2 differential gene GO enrichment analysis bubble diagram; (**c**) gene expression and GO enrichment analysis of differential genes in branch 1; (**d**) branch 1 differential gene GO enrichment analysis bubble diagram.

**Figure 10 ijms-25-00584-f010:**
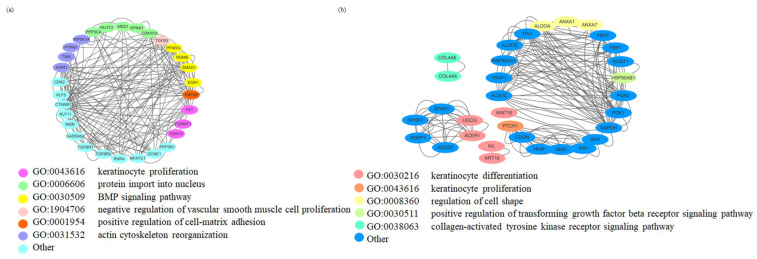
Interaction analysis of differential gene GO enrichment results during keratinocyte differentiation at different stages. (**a**) Interaction analysis of the GO enrichment results of differential genes in the differentiation process of branch 2 keratinocytes; (**b**) Interaction analysis of differential gene GO enrichment results during branch 1 keratinocyte differentiation.

**Figure 11 ijms-25-00584-f011:**
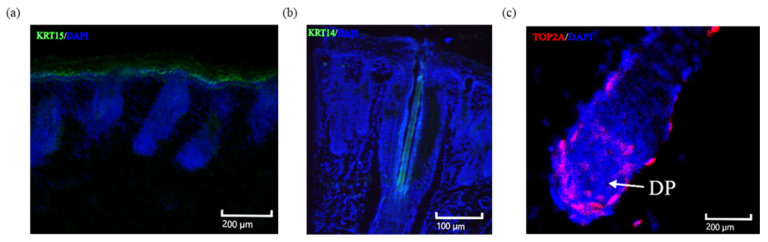
Fluorescence immunohistochemical identification of some marker genes. (**a**) KRT15 fluorescence immunohistochemical identification; (**b**) KRT14 fluorescence immunohistochemical identification; (**c**) TOP2A fluorescence immunohistochemical identification.

## Data Availability

Data will be available upon request from the corresponding author.

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
