# Peer review of "Single-Cell Transcriptome Sequence Profiling on the Morphogenesis of Secondary Hair Follicles in Ordos Fine-Wool Sheep"

_ijms, 2024, doi:10.3390/ijms25010584_

Round 1

Reviewer 1 Report

Comments and Suggestions for Authors

This manuscript is an interesting study assessing the morphogenesis of secondary hair follicles in Ordos fine-wool sheep based on single-cell transcriptome sequencing. And the authors analyzed the differential genes in primary hair follicles and secondary hair follicles. Based on the pseudo-time analysis, the differentiation trajectory of dermal lineage cells and epidermal lineage cells in the Ordos fine-wool sheep was constructed, which would provide a theoretical basis for Ordos fine-wool sheep breeding. Therefore, it could be considered to be published depending on the revision.

1.      In Figure 2, what’s kind of cell types included in the figure. Please add the name of the cell clusters.

2.      The name of the maker genes is obscure in figure 3.

3.      In figure 4, it should be “cell percentage”, not “cell numbers”.

4.      Figure 5. Comparative analysis of differential gene d in different cell populations, what does “gene d” mean?

5.      In figure 11, what stage did the authors select to validate the marker genes?

6.      Please add the details of the samples used for seRNA-seq. How many samples did the authors used in this study?

7.      The conclusions of this study are too long to read.

8.      There are many studies involved in hair follicle morphogenesis by scRNA-seq, however, this manuscript lacks in referencing them. What’s the difference or the innovation of this study compared with the previous studies?

Reviewer 2 Report

Comments and Suggestions for Authors

Dear Authors,

The proposed manuscript "Research on the morphogenesis of secondary hair follicles in Ordos fine-wool sheep based on single-cell transcriptome sequencing" is correctly written, interesting and scientifically challenging.

I would suggest a minor refinement of the title. In particular, it is not usual to use the term "research" in the title, because every scientific paper is actually research. I do not have a clear title suggestion, but perhaps it could be "Single-cell transcriptome sequence profiling on the morphogenesis of secondary hair follicles in Ordos fine-wool sheep"

The chapter "Introduction" is informative enough and introduces the reader to the research topic.

The research results are presented clearly, systematically and concisely. I noticed that the text in Figure 7 and Figure 9 is difficult to read. I suggest that the readability of the text be improved.

The "Discussion" chapter is clearly and interestingly written.

Chapter 4 "Materials and Methods" explains the research design and methods used. Some technical errors were noted (e.g., line 516 . to line 525; "Translate to professional English: After normalizing ...").

It is necessary to proofread the text.

The references in the bibliography are written correctly.
